# Guided Policy Search via Approximate Mirror Descent

**William Montgomery**
Dept. of Computer Science and Engineering
University of Washington
`wmonty@cs.washington.edu`

**Sergey Levine**
Dept. of Computer Science and Engineering
University of Washington
`svlevine@cs.washington.edu`

## Abstract

Guided policy search algorithms can be used to optimize complex nonlinear policies, such as deep neural networks, without directly computing policy gradients in the high-dimensional parameter space. Instead, these methods use supervised learning to train the policy to mimic a "teacher" algorithm, such as a trajectory optimizer or a trajectory-centric reinforcement learning method. Guided policy search methods provide asymptotic local convergence guarantees by construction, but it is not clear how much the policy improves within a small, finite number of iterations. We show that guided policy search algorithms can be interpreted as an approximate variant of mirror descent, where the projection onto the constraint manifold is not exact. We derive a new guided policy search algorithm that is simpler and provides appealing improvement and convergence guarantees in simplified convex and linear settings, and show that in the more general nonlinear setting, the error in the projection step can be bounded. We provide empirical results on several simulated robotic navigation and manipulation tasks that show that our method is stable and achieves similar or better performance when compared to prior guided policy search methods, with a simpler formulation and fewer hyperparameters.

## 1 Introduction

Policy search algorithms based on supervised learning from a computational or human "teacher" have gained prominence in recent years due to their ability to optimize complex policies for autonomous flight [16], video game playing [15, 4], and bipedal locomotion [11]. Among these methods, guided policy search algorithms [6] are particularly appealing due to their ability to adapt the teacher to produce data that is best suited for training the final policy with supervised learning. Such algorithms have been used to train complex deep neural network policies for vision-based robotic manipulation [6], as well as a variety of other tasks [19, 11]. However, convergence results for these methods typically follow by construction from their formulation as a constrained optimization, where the teacher is gradually constrained to match the learned policy, and guarantees on the performance of the final policy only hold at convergence if the constraint is enforced exactly. This is problematic in practical applications, where such algorithms are typically executed for a small number of iterations.

In this paper, we show that guided policy search algorithms can be interpreted as approximate variants of mirror descent under constraints imposed by the policy parameterization, with supervised learning corresponding to a projection onto the constraint manifold. Based on this interpretation, we can derive a new, simplified variant of guided policy search, which corresponds exactly to mirror descent under linear dynamics and convex policy spaces. When these convexity and linearity assumptions do not hold, we can show that the projection step is approximate, up to a bound that depends on the step size of the algorithm, which suggests that for a small enough step size, we can achieve continuous improvement. The form of this bound provides us with intuition about how to adjust the step size in practice, so as to obtain a simple algorithm with a small number of hyperparameters.

---
**Algorithm 1** Generic guided policy search method
---
1: **for** iteration $k \in \{1, \ldots, K\}$ **do**
2:     C-step: improve each $p_i(\mathbf{u}_t|\mathbf{x}_t)$ based on surrogate cost $\tilde{\ell}_i(\mathbf{x}_t, \mathbf{u}_t)$, return samples $\mathcal{D}_i$
3:     S-step: train $\pi_\theta(\mathbf{u}_t|\mathbf{x}_t)$ with supervised learning on the dataset $\mathcal{D} = \cup_i \mathcal{D}_i$
4:     Modify $\tilde{\ell}_i(\mathbf{x}_t, \mathbf{u}_t)$ to enforce agreement between $\pi_\theta(\mathbf{u}_t|\mathbf{x}_t)$ and each $p(\mathbf{u}_t|\mathbf{x}_t)$
5: **end for**
---

The main contribution of this paper is a simple new guided policy search algorithm that can train complex, high-dimensional policies by alternating between trajectory-centric reinforcement learning and supervised learning, as well as a connection between guided policy search methods and mirror descent. We also extend previous work on bounding policy cost in terms of KL divergence [15, 17] to derive a bound on the cost of the policy at each iteration, which provides guidance on how to adjust the step size of the method. We provide empirical results on several simulated robotic navigation and manipulation tasks that show that our method is stable and achieves similar or better performance when compared to prior guided policy search methods, with a simpler formulation and fewer hyperparameters.

## 2   Guided Policy Search Algorithms

We first review guided policy search methods and background. Policy search algorithms aim to optimize a parameterized policy $\pi_\theta(\mathbf{u}_t|\mathbf{x}_t)$ over actions $\mathbf{u}_t$ conditioned on the state $\mathbf{x}_t$. Given stochastic dynamics $p(\mathbf{x}_{t+1}|\mathbf{x}_t, \mathbf{u}_t)$ and cost $\ell(\mathbf{x}_t, \mathbf{u}_t)$, the goal is to minimize the expected cost under the policy's trajectory distribution, given by $J(\theta) = \sum_{t=1}^{T} E_{\pi_\theta(\mathbf{x}_t, \mathbf{u}_t)}[\ell(\mathbf{x}_t, \mathbf{u}_t)]$, where we overload notation to use $\pi_\theta(\mathbf{x}_t, \mathbf{u}_t)$ to denote the marginals of $\pi_\theta(\tau) = p(\mathbf{x}_1)\prod_{t=1}^{T} p(\mathbf{x}_{t+1}|\mathbf{x}_t, \mathbf{u}_t)\pi_\theta(\mathbf{u}_t|\mathbf{x}_t)$, where $\tau = \{\mathbf{x}_1, \mathbf{u}_1, \ldots, \mathbf{x}_T, \mathbf{u}_T\}$ denotes a trajectory. A standard reinforcement learning (RL) approach to policy search is to compute the gradient $\nabla_\theta J(\theta)$ and use it to improve $J(\theta)$ [18, 14]. The gradient is typically estimated using samples obtained from the real physical system being controlled, and recent work has shown that such methods can be applied to very complex, high-dimensional policies such as deep neural networks [17, 10]. However, for complex, high-dimensional policies, such methods tend to be inefficient, and practical real-world applications of such model-free policy search techniques are typically limited to policies with about one hundred parameters [3].

Instead of directly optimizing $J(\theta)$, guided policy search algorithms split the optimization into a "control phase" (which we'll call the C-step) that finds multiple simple local policies $p_i(\mathbf{u}_t|\mathbf{x}_t)$ that can solve the task from different initial states $\mathbf{x}_1^i \sim p(\mathbf{x}_1)$, and a "supervised phase" (S-step) that optimizes the global policy $\pi_\theta(\mathbf{u}_t|\mathbf{x}_t)$ to match all of these local policies using standard supervised learning. In fact, a variational formulation of guided policy search [7] corresponds to the EM algorithm, where the C-step is actually the E-step, and the S-step is the M-step. The benefit of this approach is that the local policies $p_i(\mathbf{u}_t|\mathbf{x}_t)$ can be optimized separately using domain-specific local methods. Trajectory optimization might be used when the dynamics are known [19, 11], while local RL methods might be used with unknown dynamics [5, 6], which still requires samples from the real system, though substantially fewer than the direct approach, due to the simplicity of the local policies. This sample efficiency is the main advantage of guided policy search, which can train policies with nearly a hundred thousand parameters for vision-based control using under 200 episodes [6], in contrast to direct deep RL methods that might require orders of magnitude more experience [17, 10].

A generic guided policy search method is shown in Algorithm 1. The C-step invokes a local policy optimizer (trajectory optimization or local RL) for each $p_i(\mathbf{u}_t|\mathbf{x}_t)$ on line 2, and the S-step uses supervised learning to optimize the global policy $\pi_\theta(\mathbf{u}_t|\mathbf{x}_t)$ on line 3 using samples from each $p_i(\mathbf{u}_t|\mathbf{x}_t)$, which are generated during the C-step. On line 4, the surrogate cost $\tilde{\ell}_i(\mathbf{x}_t, \mathbf{u}_t)$ for each $p_i(\mathbf{u}_t|\mathbf{x}_t)$ is adjusted to ensure convergence. This step is crucial, because supervised learning does not in general guarantee that $\pi_\theta(\mathbf{u}_t|\mathbf{x}_t)$ will achieve similar long-horizon performance to $p_i(\mathbf{u}_t|\mathbf{x}_t)$ [15]. The local policies might not even be reproducible by a single global policy in general. To address this issue, most guided policy search methods have some mechanism to force the local policies to agree with the global policy, typically by framing the entire algorithm as a constrained optimization that seeks at convergence to enforce equality between $\pi_\theta(\mathbf{u}_t|\mathbf{x}_t)$ and each $p_i(\mathbf{u}_t|\mathbf{x}_t)$. The form of the

overall optimization problem resembles dual decomposition, and usually looks something like this:

$$\min_{\theta,p_1,\ldots,p_N} \sum_{i=1}^{N}\sum_{t=1}^{T} E_{p_i(\mathbf{x}_t,\mathbf{u}_t)}[\ell(\mathbf{x}_t,\mathbf{u}_t)] \text{ such that } p_i(\mathbf{u}_t|\mathbf{x}_t) = \pi_\theta(\mathbf{u}_t|\mathbf{x}_t) \ \forall \mathbf{x}_t,\mathbf{u}_t,t,i. \tag{1}$$

Since $\mathbf{x}_1^i \sim p(\mathbf{x}_1)$, we have $J(\theta) \approx \sum_{i=1}^{N}\sum_{t=1}^{T} E_{p_i(\mathbf{x}_t,\mathbf{u}_t)}[\ell(\mathbf{x}_t,\mathbf{u}_t)]$ when the constraints are enforced exactly. The particular form of the constraint varies depending on the method: prior works have used dual gradient descent [8], penalty methods [11], ADMM [12], and Bregman ADMM [6]. We omit the derivation of these prior variants due to space constraints.

## 2.1 Efficiently Optimizing Local Policies

A common and simple choice for the local policies $p_i(\mathbf{u}_t|\mathbf{x}_t)$ is to use time-varying linear-Gaussian controllers of the form $p_i(\mathbf{u}_t|\mathbf{x}_t) = \mathcal{N}(\mathbf{K}_t\mathbf{x}_t + \mathbf{k}_t, \mathbf{C}_t)$, though other options are also possible [12, 11, 19]. Linear-Gaussian controllers represent individual trajectories with linear stabilization and Gaussian noise, and are convenient in domains where each local policy can be trained from a different (but consistent) initial state $\mathbf{x}_1^i \sim p(\mathbf{x}_1)$. This represents an additional assumption beyond standard RL, but allows for an extremely efficient and convenient local model-based RL algorithm based on iterative LQR [9]. The algorithm proceeds by generating $N$ samples on the real physical system from each local policy $p_i(\mathbf{u}_t|\mathbf{x}_t)$ during the C-step, using these samples to fit local linear-Gaussian dynamics for each local policy of the form $p_i(\mathbf{x}_{t+1}|\mathbf{x}_t,\mathbf{u}_t) = \mathcal{N}(f_{\mathbf{x}t}\mathbf{x}_t + f_{\mathbf{u}t}\mathbf{u}_t + f_{ct}, \mathbf{F}_t)$ using linear regression, and then using these fitted dynamics to improve the linear-Gaussian controller via a modified LQR algorithm [5]. This modified LQR method solves the following optimization problem:

$$\min_{\mathbf{K}_t,\mathbf{k}_t,\mathbf{C}_t} \sum_{t=1}^{T} E_{p_i(\mathbf{x}_t,\mathbf{u}_t)}[\tilde{\ell}_i(\mathbf{x}_t,\mathbf{u}_t)] \text{ such that } D_{\mathrm{KL}}(p_i(\tau)\|\bar{p}_i(\tau)) \le \epsilon, \tag{2}$$

where we again use $p_i(\tau)$ to denote the trajectory distribution induced by $p_i(\mathbf{u}_t|\mathbf{x}_t)$ and the fitted dynamics $p_i(\mathbf{x}_{t+1}|\mathbf{x}_t,\mathbf{u}_t)$. Here, $\bar{p}_i(\mathbf{u}_t|\mathbf{x}_t)$ denotes the previous local policy, and the constraint ensures that the change in the local policy is bounded, as proposed also in prior works [1, 14, 13]. This is particularly important when using linearized dynamics fitted to local samples, since these dynamics are not valid outside of a small region around the current controller. In the case of linear-Gaussian dynamics and policies, the KL-divergence constraint $D_{\mathrm{KL}}(p_i(\tau)\|\bar{p}_i(\tau)) \le \epsilon$ can be shown to simplify, as shown in prior work [5] and Appendix A:

$$D_{\mathrm{KL}}(p_i(\tau)\|\bar{p}_i(\tau)) = \sum_{t=1}^{T} D_{\mathrm{KL}}(p_i(\mathbf{u}_t|\mathbf{x}_t)\|\bar{p}_i(\mathbf{u}_t|\mathbf{x}_t)) = \sum_{t=1}^{T} -E_{p_i(\mathbf{x}_t,\mathbf{u}_t)}[\log\bar{p}_i(\mathbf{u}_t|\mathbf{x}_t)] - \mathcal{H}(p_i(\mathbf{u}_t|\mathbf{x}_t)),$$

and the resulting Lagrangian of the problem in Equation (2) can be optimized with respect to the primal variables using the standard LQR algorithm, which suggests a simple method for solving the problem in Equation (2) using dual gradient descent [5]. The surrogate objective $\tilde{\ell}_i(\mathbf{x}_t,\mathbf{u}_t) = \ell(\mathbf{x}_t,\mathbf{u}_t) + \phi_i(\theta)$ typically includes some term $\phi_i(\theta)$ that encourages the local policy $p_i(\mathbf{u}_t|\mathbf{x}_t)$ to stay close to the global policy $\pi_\theta(\mathbf{u}_t|\mathbf{x}_t)$, such as a KL-divergence of the form $D_{\mathrm{KL}}(p_i(\mathbf{u}_t|\mathbf{x}_t)\|\pi_\theta(\mathbf{u}_t|\mathbf{x}_t))$.

## 2.2 Prior Convergence Results

Prior work on guided policy search typically shows convergence by construction, by framing the C-step and S-step as block coordinate ascent on the (augmented) Lagrangian of the problem in Equation (1), with the surrogate cost $\tilde{\ell}_i(\mathbf{x}_t,\mathbf{u}_t)$ for the local policies corresponding to the (augmented) Lagrangian, and the overall algorithm being an instance of dual gradient descent [8], ADMM [12], or Bregman ADMM [6]. Since these methods enforce the constraint $p_i(\mathbf{u}_t|\mathbf{x}_t) = \pi_\theta(\mathbf{u}_t|\mathbf{x}_t)$ at convergence (up to linearization or sampling error, depending on the method), we know that $\frac{1}{N}\sum_{i=1}^{N} E_{p_i(\mathbf{x}_t,\mathbf{u}_t)}[\ell(\mathbf{x}_t,\mathbf{u}_t)] \approx E_{\pi_\theta(\mathbf{x}_t,\mathbf{u}_t)}[\ell(\mathbf{x}_t,\mathbf{u}_t)]$ at convergence.[1] However, prior work does not say anything about $\pi_\theta(\mathbf{u}_t|\mathbf{x}_t)$ at intermediate iterations, and the constraints of policy search in the real world might often preclude running the method to full convergence. We propose a simplified variant of guided policy search, and present an analysis that sheds light on the performance of both the new algorithm and prior guided policy search methods.

**Algorithm 2** Mirror descent guided policy search (MDGPS): convex linear variant

1: **for** iteration $k \in \{1, \ldots, K\}$ **do**
2:     C-step: $p_i \leftarrow \arg\min_{p_i} E_{p_i(\tau)} \left[ \sum_{t=1}^{T} \ell(\mathbf{x}_t, \mathbf{u}_t) \right]$ such that $D_{\mathrm{KL}}(p_i(\tau) \| \pi_\theta(\tau)) \le \epsilon$
3:     S-step: $\pi_\theta \leftarrow \arg\min_\theta \sum_i D_{\mathrm{KL}}(p_i(\tau) \| \pi_\theta(\tau))$ (via supervised learning)
4: **end for**

---

## 3   Mirror Descent Guided Policy Search

In this section, we propose our new simplified guided policy search, which we term mirror descent guided policy search (MDGPS). This algorithm uses the constrained LQR optimization in Equation (2) to optimize each of the local policies, but instead of constraining each local policy $p_i(\mathbf{u}_t | \mathbf{x}_t)$ against the previous local policy $\bar{p}_i(\mathbf{u}_t | \mathbf{x}_t)$, we instead constraint it directly against the global policy $\pi_\theta(\mathbf{u}_t | \mathbf{x}_t)$, and simply set the surrogate cost to be the true cost, such that $\tilde{\ell}_i(\mathbf{x}_t, \mathbf{u}_t) = \ell(\mathbf{x}_t, \mathbf{u}_t)$. The method is summarized in Algorithm 2. In the case of linear dynamics and a quadratic cost (i.e. the LQR setting), and assuming that supervised learning can globally solve a convex optimization problem, we can show that this method corresponds to an instance of mirror descent [2] on the objective $J(\theta)$. In this formulation, the optimization is performed on the space of trajectory distributions, with a constraint that the policy must lie on the manifold of policies with the chosen parameterization. Let $\Pi_\Theta$ be the set of all possible policies $\pi_\theta$ for a given parameterization, where we overload notation to also let $\Pi_\Theta$ denote the set of trajectory distributions that are possible under the chosen parameterization. The return $J(\theta)$ can be optimized according to $\pi_\theta \leftarrow \arg\min_{\pi \in \Pi_\Theta} E_{\pi(\tau)}[\sum_{t=1}^{T} \ell(\mathbf{x}_t, \mathbf{u}_t)]$. Mirror descent solves this optimization by alternating between two steps at each iteration $k$:

$$ p^k \leftarrow \arg\min_p E_{p(\tau)} \left[ \sum_{t=1}^{T} \ell(\mathbf{x}_t, \mathbf{u}_t) \right] \text{ s. t. } D\left(p, \pi^k\right) \le \epsilon, \qquad \pi^{k+1} \leftarrow \arg\min_{\pi \in \Pi_\Theta} D\left(p^k, \pi\right). $$

The first step finds a new distribution $p^k$ that minimizes the cost and is close to the previous policy $\pi^k$ in terms of the divergence $D\left(p, \pi^k\right)$, while the second step projects this distribution onto the constraint set $\Pi_\Theta$, with respect to the divergence $D(p^k, \pi)$. In the linear-quadratic case with a convex supervised learning phase, this corresponds exactly to Algorithm 2: the C-step optimizes $p^k$, while the S-step is the projection. Monotonic improvement of the global policy $\pi_\theta$ follows from the monotonic improvement of mirror descent [2]. In the case of linear-Gaussian dynamics and policies, the S-step, which minimizes KL-divergence between trajectory distributions, in fact only requires minimizing the KL-divergence between policies. Using the identity in Appendix A, we know that

$$ D_{\mathrm{KL}}(p_i(\tau) \| \pi_\theta(\tau)) = \sum_{t=1}^{T} E_{p_i(\mathbf{x}_t)} \left[ D_{\mathrm{KL}}(p_i(\mathbf{u}_t | \mathbf{x}_t) \| \pi_\theta(\mathbf{u}_t | \mathbf{x}_t)) \right]. \tag{3} $$

### 3.1   Implementation for Nonlinear Global Policies and Unknown Dynamics

In practice, we aim to optimize complex policies for nonlinear systems with unknown dynamics. This requires a few practical considerations. The C-step requires a local quadratic cost function, which can be obtained via Taylor expansion, as well as local linear-Gaussian dynamics $p(\mathbf{x}_{t+1} | \mathbf{x}_t, \mathbf{u}_t) = \mathcal{N}(f_{\mathbf{x}t}\mathbf{x}_t + f_{\mathbf{u}t}\mathbf{u}_t + f_{ct}, \mathbf{F}_t)$, which we can fit to samples as in prior work [5]. We also need a local time-varying linear-Gaussian approximation to the global policy $\pi_\theta(\mathbf{u}_t | \mathbf{x}_t)$, denoted $\bar{\pi}_{\theta i}(\mathbf{u}_t | \mathbf{x}_t)$. This can be obtained either by analytically differentiating the policy, or by using the same linear regression method that we use to estimate $p(\mathbf{x}_{t+1} | \mathbf{x}_t, \mathbf{u}_t)$, which is the approach in our implementation. In both cases, we get a different global policy linearization around each local policy. Following prior work [5], we use a Gaussian mixture model prior for both the dynamics and global policy fit.

The S-step can be performed approximately in the nonlinear case by using the samples collected for dynamics fitting to also train the global policy. Following prior work [6], our S-step minimizes[2]

$$ \sum_{i,t} E_{p_i(\mathbf{x}_t)} \left[ D_{\mathrm{KL}}(\pi_\theta(\mathbf{u}_t | \mathbf{x}_t) \| p_i(\mathbf{u}_t | \mathbf{x}_t)) \right] \approx \frac{1}{|\mathcal{D}_i|} \sum_{i,t,j} D_{\mathrm{KL}}(\pi_\theta(\mathbf{u}_t | \mathbf{x}_{t,i,j}) \| p_i(\mathbf{u}_t | \mathbf{x}_{t,i,j})), $$

**Algorithm 3** Mirror descent guided policy search (MDGPS): unknown nonlinear dynamics

---
1: **for** iteration $k \in \{1, \ldots, K\}$ **do**
2:     Generate samples $\mathcal{D}_i = \{\tau_{i,j}\}$ by running either $p_i$ or $\pi_{\theta i}$
3:     Fit linear-Gaussian dynamics $p_i(\mathbf{x}_{t+1}|\mathbf{x}_t, \mathbf{u}_t)$ using samples in $\mathcal{D}_i$
4:     Fit linearized global policy $\bar{\pi}_{\theta i}(\mathbf{u}_t|\mathbf{x}_t)$ using samples in $\mathcal{D}_i$
5:     C-step: $p_i \leftarrow \arg\min_{p_i} E_{p_i(\tau)}[\sum_{t=1}^{T} \ell(\mathbf{x}_t, \mathbf{u}_t)]$ such that $D_{\mathrm{KL}}(p_i(\tau)\|\bar{\pi}_{\theta i}(\tau)) \leq \epsilon$
6:     S-step: $\pi_\theta \leftarrow \arg\min_\theta \sum_{t,i,j} D_{\mathrm{KL}}(\pi_\theta(\mathbf{u}_t|\mathbf{x}_{t,i,j})\|p_i(\mathbf{u}_t|\mathbf{x}_{t,i,j}))$ (via supervised learning)
7:     Adjust $\epsilon$ (see Section 4.2)
8: **end for**

---

where $\mathbf{x}_{t,i,j}$ is the $j^{\text{th}}$ sample from $p_i(\mathbf{x}_t)$ obtained by running $p_i(\mathbf{u}_t|\mathbf{x}_t)$ on the real system. For linear-Gaussian $p_i(\mathbf{u}_t|\mathbf{x}_t)$ and (nonlinear) conditionally Gaussian $\pi_\theta(\mathbf{u}_t|\mathbf{x}_t) = \mathcal{N}(\mu^\pi(\mathbf{x}_t), \Sigma^\pi(\mathbf{x}_t))$, where $\mu^\pi$ and $\Sigma^\pi$ can be any function (such as a deep neural network), the KL-divergence $D_{\mathrm{KL}}(\pi_\theta(\mathbf{u}_t|\mathbf{x}_{t,i,j})\|p_i(\mathbf{u}_t|\mathbf{x}_{t,i,j}))$ can easily be evaluated and differentiated in closed form [6]. However, in the nonlinear setting, minimizing this objective no longer minimizes the KL-divergence between trajectory distributions $D_{\mathrm{KL}}(\pi_\theta(\tau)\|p_i(\tau))$ exactly, which means that MDGPS does not correspond exactly to mirror descent: although the C-step can still be evaluated exactly, the S-step now corresponds to an approximate projection onto the constraint manifold. In the next section, we discuss how we can bound the error in this projection. A summary of the nonlinear MDGPS method is provided in Algorithm 4, and additional details are in Appendix B. The samples for linearizing the dynamics and policy can be obtained by running either the last local policy $p_i(\mathbf{u}_t|\mathbf{x}_t)$, or the last global policy $\pi_\theta(\mathbf{u}_t|\mathbf{x}_t)$. Both variants produce good results, and we compare them in Section 6.

### 3.2 Analysis of Prior Guided Policy Search Methods as Approximate Mirror Descent

The main distinction between the proposed method and prior guided policy search methods is that the constraint $D_{\mathrm{KL}}(p_i(\tau)\|\bar{\pi}_{\theta i}(\tau)) \leq \epsilon$ is enforced on the local policies at each iteration, while in prior methods, this constraint is iteratively enforced via a dual descent procedure over multiple iterations. This means that the prior methods perform approximate mirror descent with step sizes that are adapted (by adjusting the Lagrange multipliers) but not constrained exactly. In our empirical evaluation, we show that our approach is somewhat more stable, though sometimes slower than these prior methods. This empirical observation agrees with our intuition: prior methods can sometimes be faster, because they do not exactly constrain the step size, but our method is simpler, requires less tuning, and always takes bounded steps on the global policy in trajectory space.

## 4 Analysis in the Nonlinear Case

Although the S-step under nonlinear dynamics is not an optimal projection onto the constraint manifold, we can bound the additional cost incurred by this projection in terms of the KL-divergence between $p_i(\mathbf{u}_t|\mathbf{x}_t)$ and $\pi_\theta(\mathbf{u}_t|\mathbf{x}_t)$. This analysis also reveals why prior guided policy search algorithms, which only have asymptotic convergence guarantees, still attain good performance in practice even after a small number of iterations. We will drop the subscript $i$ from $p_i(\mathbf{u}_t|\mathbf{x}_t)$ in this section for conciseness, though the same analysis can be repeated for multiple local policies $p_i(\mathbf{u}_t|\mathbf{x}_t)$.

### 4.1 Bounding the Global Policy Cost

The analysis in this section is based on the following lemma, which we prove in Appendix C.1, building off of earlier results by Ross et al. [15] and Schulman et al. [17]:

**Lemma 4.1** *Let $\epsilon_t = \max_{\mathbf{x}_t} D_{KL}(p(\mathbf{u}_t|\mathbf{x}_t)\|\pi_\theta(\mathbf{u}_t|\mathbf{x}_t))$. Then $D_{TV}(p(\mathbf{x}_t)\|\pi_\theta(\mathbf{x}_t)) \leq 2\sum_{t=1}^{T} \sqrt{2\epsilon_t}$.*

This means that if we can bound the KL-divergence between the policies, then the total variation divergence between their state marginals (given by $D_{\mathrm{TV}}(p(\mathbf{x}_t)\|\pi_\theta(\mathbf{x}_t)) = \frac{1}{2}\|p(\mathbf{x}_t) - \pi_\theta(\mathbf{x}_t)\|_1$) will also be bounded. This bound allows us in turn to relate the total expected costs of the two policies to each other according to the following lemma, which we prove in Appendix C.2:

**Lemma 4.2** *If $D_{TV}(p(\mathbf{x}_t)\|\pi_\theta(\mathbf{x}_t)) \le 2\sum_{t=1}^T \sqrt{2\epsilon_t}$, then we can bound the total cost of $\pi_\theta$ as*

$$\sum_{t=1}^T E_{\pi_\theta(\mathbf{x}_t,\mathbf{u}_t)}[\ell(\mathbf{x}_t,\mathbf{u}_t)] \le \sum_{t=1}^T \left[ E_{p(\mathbf{x}_t,\mathbf{u}_t)}[\ell(\mathbf{x}_t,\mathbf{u}_t)] + \sqrt{2\epsilon_t} \max_{\mathbf{x}_t,\mathbf{u}_t} \ell(\mathbf{x}_t,\mathbf{u}_t) + 2\sqrt{2\epsilon_t}Q_{max,t} \right]$$

*where $Q_{max,t} = \sum_{t'=t}^T \max_{\mathbf{x}_{t'},\mathbf{u}_{t'}} \ell(\mathbf{x}_{t'},\mathbf{u}_{t'})$, the maximum total cost from time $t$ to $T$.*

This bound on the cost of $\pi_\theta(\mathbf{u}_t|\mathbf{x}_t)$ tells us that if we update $p(\mathbf{u}_t|\mathbf{x}_t)$ so as to decrease its total cost or decrease its KL-divergence against $\pi_\theta(\mathbf{u}_t|\mathbf{x}_t)$, we will eventually reduce the cost of $\pi_\theta(\mathbf{u}_t|\mathbf{x}_t)$. For the MDGPS algorithm, this bound suggests that we can ensure improvement of the global policy within a small number of iterations by appropriately choosing the constraint $\epsilon$ during the C-step. Recall that the C-step constrains $\sum_{t=1}^T \epsilon_t \le \epsilon$, so if we choose $\epsilon$ to be small enough, we can close the gap between the local and global policies. Optimizing the bound directly turns out to produce very slow learning in practice, because the bound is very loose. However, it tells us that we can either decrease $\epsilon$ toward the end of the optimization process or if we observe the global policy performing much worse than the local policies. We discuss how this idea can be put into action in the next section.

## 4.2 Step Size Selection

Setting the local policy step size $\epsilon$ is important for proper convergence of guided policy search methods. Since we are approximating the true unknown dynamics with time-varying linear dynamics, setting $\epsilon$ too large can produce unstable local policies which cause the method to fail. However, setting $\epsilon$ too small will prevent the local policies from improving significantly between iterations, leading to slower learning rates.

In prior work [8], the step size $\epsilon$ in the local policy optimization is dynamically adjusted by considering the difference between the predicted change in the cost of the local policy $p(\mathbf{u}_t|\mathbf{x}_t)$ under the fitted dynamics, and the actual cost obtained when sampling from that policy. The intuition is that, because the linearized dynamics are local, we incur a larger cost the further we deviate from the previous policy. We can adjust the step size by estimating the rate at which the additional cost is incurred and choosing the optimal tradeoff. In Appendix B.3 we describe the step size adjustment rule used for BADMM in prior work, and use it to derive two step size adjustment rules for MDGPS: "classic" and "global." The classic step size adjustment is a direct reintrepretation of the BADMM step rule for MDGPS, while the global step rule is a more conservative rule that takes the difference between the global and local policies into account.

## 5 Relation to Prior Work

While we've discussed the connections between MDGPS and prior guided policy search methods, in this section we'll also discuss the connections between our method and other policy search methods. One popular supervised policy learning methods is DAGGER [15], which also trains the policy using supervised learning, but does not attempt to adapt the teacher to provide better training data. MDGPS removes the assumption in DAGGER that the supervised learning stage has bounded error against an arbitrary teacher policy. MDGPS does not need to make this assumption, since the teacher can be adapted to the limitations of the global policy learning. This is particularly important when the global policy has computational or observational limitations, such as when learning to use camera images for partially observed control tasks or, as shown in our evaluation, blind peg insertion.

When we sample from the global policy $\pi_\theta(\mathbf{u}_t|\mathbf{x}_t)$, our method resembles policy gradient methods with KL-divergence constraints [14, 13, 17]. However, policy gradient methods update the policy $\pi_\theta(\mathbf{u}_t|\mathbf{x}_t)$ at each iteration by linearizing with respect to the policy parameters, which often requires small steps for complex, nonlinear policies, such as neural networks. In contrast, we linearize in the space of time-varying linear dynamics, while the policy is optimized at each iteration with many steps of supervised learning (e.g. stochastic gradient descent). This makes MDGPS much better suited for quickly and efficiently training highly nonlinear, high-dimensional policies.

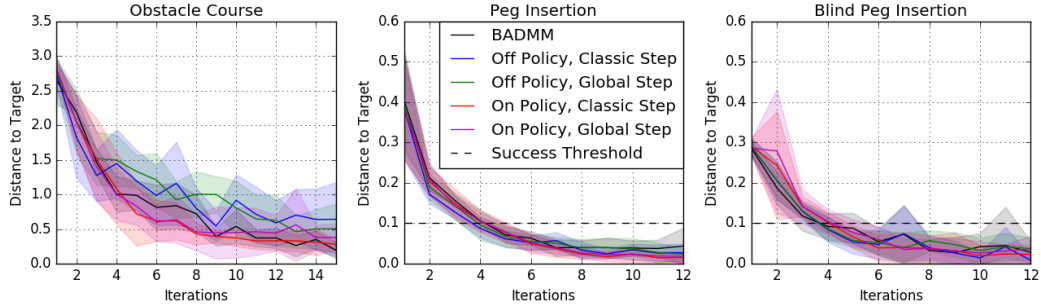

Figure 1: Results for MDGPS variants and BADMM on each task. MDGPS is tested with local policy ("off policy") and global policy ("on policy") sampling (see Section 3.1), and both the "classic" and "global" step sizes (see Section 4.2). The vertical axis for the obstacle task shows the average distance between the point mass and the target. The vertical axis for the peg tasks shows the average distance between the bottom of the peg and the hole. Distances above 0.1, which is the depth of the hole (shown as a dotted line) indicate failure. All experiments are repeated ten times, with the average performance and standard deviation shown in the plots.

## 6 Experimental Evaluation

We compare several variants of MDGPS and a prior guided policy search method based on Bregman ADMM (BADMM) [6]. We evaluate all methods on one simulated robotic navigation task and two manipulation tasks. For MDGPS, during training we sample from either the local policies ("off-policy" sampling) or the global policy ("on-policy" sampling), and we use both forms of the step rule described in Section 4.2 ("classic" and "global"). [3]

**Obstacle Navigation.** In this task, a 2D point mass (grey) must navigate around obstacles to reach a target (shown in green), using velocities and positions relative to the target. We use $N = 5$ initial states, with 5 samples per initial state per iteration. The target and obstacles are fixed, but the starting position varies.

**Peg Insertion.** This task, which is more complex, requires controlling a 7 DoF 3D arm to insert a tight-fitting peg into a hole. The hole can be in different positions, and the state consists of joint angles, velocities, and end-effector positions relative to the target. This task is substantially more challenging physically. We use $N = 9$ different hole positions, with 5 samples per initial state per iteration.

**Blind Peg Insertion.** The last task is a blind variant of the peg insertion task, where the target-relative end effector positions are provided to the local policies, but not to the global policy $\pi_\theta(\mathbf{u}_t|\mathbf{x}_t)$. This requires the global policy to search for the hole, since no input to the global policy can distinguish between the different initial state $\mathbf{x}_1^i$. This makes it much more challenging to adapt the global and local policies to each other, and makes it impossible for the global learner to succeed without adaptation of the local policies. We use $N = 4$ different hole positions, with 5 samples per initial state per iteration.

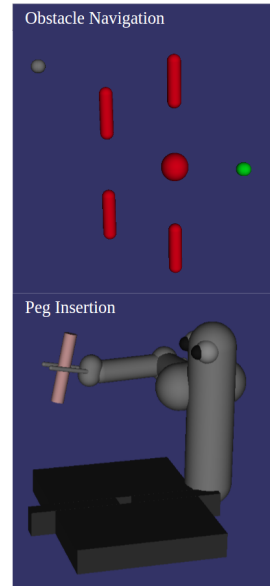

The global policy for each task consists of a fully connected neural network with two hidden layers with 40 rectified linear units. The same settings are used for MDGPS and the prior BADMM-based method, except for the difference in surrogate costs, constraints, and step size adjustment methods discussed in the paper. Results are presented in Figure 1 and Table 1. On the easier point mass navigation task all methods achieve similar performance, but the on-policy variants of MDGPS outperform the off-policy variants. This suggests that we can benefit from directly sampling from the global policy during training, which is not possible in the BADMM formulation. Although performance is similar among all methods, the MDGPS methods are all substantially easier to apply to these tasks, since they have very few free hyperparameters. An initial step size must be selected, but the adaptive step size adjustment rules make this choice less important. In contrast,

|  | Itr. | BADMM | Off/Classic | Off/Global | On/Classic | On/Global |
|---|---|---|---|---|---|---|
| Peg | 3 | $1.1\% \pm 3.3\%$ | $\mathbf{11.1 \pm 9.9\%}$ | $6.7\% \pm 7.4\%$ | $6.7\% \pm 7.4\%$ | $6.7\% \pm 7.4\%$ |
| Peg | 6 | $51.1\% \pm 10.2\%$ | $62.2 \pm 17.4\%$ | $64.4\% \pm 19.1\%$ | $\mathbf{68.9\% \pm 18.5\%}$ | $63.3\% \pm 20.0\%$ |
| Peg | 9 | $72.2\% \pm 14.3\%$ | $82.2 \pm 11.3\%$ | $71.1\% \pm 24.0\%$ | $\mathbf{90.0\% \pm 10.5\%}$ | $85.6\% \pm 8.7\%$ |
| Peg | 12 | $74.4\% \pm 19.3\%$ | $83.3 \pm 11.4\%$ | $84.4\% \pm 15.1\%$ | $\mathbf{90.0\% \pm 11.6\%}$ | $87.8\% \pm 13.6\%$ |
| Blind Peg | 3 | $\mathbf{20.0\% \pm 31.2\%}$ | $2.5 \pm 7.5\%$ | $7.5\% \pm 16.0\%$ | $2.5\% \pm 7.5\%$ | $15.0\% \pm 30.0\%$ |
| Blind Peg | 6 | $65.0\% \pm 22.9\%$ | $62.5 \pm 32.1\%$ | $70.0\% \pm 21.8\%$ | $\mathbf{72.5\% \pm 28.4\%}$ | $70.0\% \pm 35.0\%$ |
| Blind Peg | 9 | $82.5\% \pm 25.1\%$ | $80.0 \pm 24.5\%$ | $60.0\% \pm 32.0\%$ | $80.0\% \pm 35.0\%$ | $\mathbf{82.5\% \pm 19.5\%}$ |
| Blind Peg | 12 | $82.5\% \pm 16.1\%$ | $\mathbf{95.0 \pm 10.0\%}$ | $85.0\% \pm 22.9\%$ | $85.0\% \pm 20.0\%$ | $85.0\% \pm 12.2\%$ |

Table 1: Success rates of each method on each peg insertion task. Success is defined as inserting the peg into the hole with a final distance of less than 0.06. Results are averaged over ten runs.

the BADMM method requires choosing an initial weight on the augmented Lagrangian term, an adjustment schedule for this term, a step size on the dual variables, and a step size for local policies, all of which have a substantial impact on the final performance of the method (the reported results are for the best setting of these parameters, identified with a hyperparameter sweep).

On the peg insertion tasks, all variants MDGPS consistently outperform BADMM as shown by the success rates in Table 1, which shows that the MDGPS policies succeed at actually inserting the peg into the hole more often and on more conditions. This suggests that our method is better able to improve global policies, particularly in situations where informational or representational constraints make naïve imitation of the local policies insufficient to solve the task. On both tasks, we see faster learning from the on-policy variants, although this is less noticeable on the harder blind peg insertion task, where the best final policy is the off-policy variant with classic step size adjustment. Sampling from the global policies may be desirable in practice, since the global policies can directly use observations at runtime instead of requiring access to the state [6]. The global step size also tends to be more conservative than the classic step size, but produces more consistent and monotonic improvement.

# 7 Discussion and Future Work

We presented a new guided policy search method that corresponds to mirror descent under linearity and convexity assumptions, and showed how prior guided policy search methods can be seen as approximating mirror descent. We provide a bound on the return of the global policy in the nonlinear case, and argue that an appropriate step size can provide improvement of the global policy in this case also. Our analysis provides us with the intuition to design an automated step size adjustment rule, and we illustrate empirically that our method achieves good results on a complex simulated robotic manipulation task while requiring substantially less tuning and hyperparameter optimization than prior guided policy search methods. Manual tuning and hyperparameter searches are a major challenge across a range of deep reinforcement learning algorithms, and developing scalable policy search methods that are simple and reliable is vital to enable further progress.

As discussed in Section 5, MDGPS has interesting connections to other policy search methods. Like DAGGER [15], MDGPS uses supervised learning to train the policy, but unlike DAGGER, MDGPS does not assume that the learner is able to reproduce an arbitrary teacher's behavior with bounded error, which makes it very appealing for tasks with partial observability or other limits on information, such as learning to use camera images for robotic manipulation [6]. When sampling directly from the global policy, MDGPS also has close connections to policy gradient methods that take steps of fixed KL-divergence [14, 17], but with the steps taken in the space of trajectories rather than policy parameters, followed by a projection step. In future work, it would be interesting to explore this connection further, so as to develop new model-free policy gradient methods.

**Acknowledgments**

We thank the anonymous reviewers for their helpful and constructive feedback. This research was supported in part by an ONR Young Investigator Program award.

## Footnotes

[1] As mentioned previously, the initial state $\mathbf{x}_1^i$ of each local policy $p_i(\mathbf{u}_t|\mathbf{x}_t)$ is assumed to be drawn from $p(\mathbf{x}_1)$, hence the outer sum corresponds to Monte Carlo integration of the expectation under $p(\mathbf{x}_1)$.

[2] Note that we flip the KL-divergence inside the expectation, following [6]. We found that this produced better results. The intuition behind this is that, because $\log p_i(\mathbf{u}_t | \mathbf{x}_t)$ is proportional to the Q-function of $p_i(\mathbf{u}_t | \mathbf{x}_t)$ (see Appendix B.1), $D_{\mathrm{KL}}(\pi_\theta(\mathbf{u}_t | \mathbf{x}_{t,i,j}) \| p_i(\mathbf{u}_t | \mathbf{x}_{t,i,j}))$ minimizes the cost-to-go under $p_i(\mathbf{u}_t | \mathbf{x}_t)$ with respect to $\pi_\theta(\mathbf{u}_t | \mathbf{x}_t)$, which provides for a more informative objective than the unweighted likelihood in Equation (3).

[3] Guided policy search code, including BADMM and MDGPS methods, is available at https://www.github.com/cbfinn/gps.

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
