[Supplementary Material]

## A  KL Divergence Between Gaussian Trajectory Distributions

In this appendix, we derive the KL-divergence between two Gaussian trajectory distributions corresponding to time-varying linear-Gaussian dynamics $p(\mathbf{x}_{t+1}|\mathbf{x}_t, \mathbf{u}_t)$ and two policies $p(\mathbf{u}_t|\mathbf{x}_t)$ and $q(\mathbf{u}_t|\mathbf{x}_t)$. The two policies induce Gaussian trajectory distributions (with block-diagonal covariances) according to

$$p(\tau) = p(\mathbf{x}_1)\prod_{t=1}^{T} p(\mathbf{x}_{t+1}|\mathbf{x}_t, \mathbf{u}_t)p(\mathbf{u}_t|\mathbf{x}_t), \quad q(\tau) = p(\mathbf{x}_1)\prod_{t=1}^{T} p(\mathbf{x}_{t+1}|\mathbf{x}_t, \mathbf{u}_t)q(\mathbf{u}_t|\mathbf{x}_t).$$

We can therefore derive their KL-divergence as

$$D_{\text{KL}}(p(\tau)\|q(\tau)) = E_{p(\tau)}\left[\log p(\tau) - \log q(\tau)\right]$$

$$= E_{p(\tau)}\left[\sum_{t=1}^{T} \log p(\mathbf{u}_t|\mathbf{x}_t) - \log q(\mathbf{u}_t|\mathbf{x}_t)\right]$$

$$= \sum_{t=1}^{T} E_{p(\mathbf{x}_t, \mathbf{u}_t)}\left[\log p(\mathbf{u}_t|\mathbf{x}_t) - \log q(\mathbf{u}_t|\mathbf{x}_t)\right]$$

$$= \sum_{t=1}^{T} -E_{p(\mathbf{x}_t, \mathbf{u}_t)}\left[\log q(\mathbf{u}_t|\mathbf{x}_t)\right] - E_{p(\mathbf{x}_t)}[\mathcal{H}(p(\mathbf{u}_t|\mathbf{x}_t))]$$

$$= \sum_{t=1}^{T} -E_{p(\mathbf{x}_t, \mathbf{u}_t)}\left[\log q(\mathbf{u}_t|\mathbf{x}_t)\right] - \mathcal{H}(p(\mathbf{u}_t|\mathbf{x}_t))$$

where the second step follows because the dynamics and initial state distribution cancel, the third step follows by linearity of expectations, the fourth step from the definition of differential entropy, and the last step follows from the fact that the entropy of a conditional Gaussian distribution is independent on the quantity that it is conditioned on, since it depends only on the covariance and not the mean. We therefore have

$$D_{\text{KL}}(p(\tau)\|q(\tau)) = \sum_{t=1}^{T} -E_{p(\mathbf{x}_t, \mathbf{u}_t)}\left[\log q(\mathbf{u}_t|\mathbf{x}_t)\right] - \mathcal{H}(p(\mathbf{u}_t|\mathbf{x}_t)).$$

By the definition of KL-divergence, we can also write this as

$$D_{\text{KL}}(p(\tau)\|q(\tau)) = \sum_{t=1}^{T} E_{p(\mathbf{x}_t, \mathbf{u}_t)}\left[D_{\text{KL}}(p(\mathbf{u}_t|\mathbf{x}_t)\|q(\mathbf{u}_t\|\mathbf{x}_t))\right].$$

## B  Details of the MDGPS Algorithm

A summary of the MDGPS algorithm appears in Algorithm 2, and is repeated below for convenience:

---
**Algorithm 4** Mirror descent guided policy search (MDGPS): unknown nonlinear dynamics
---
1: **for** iteration $k \in \{1, \ldots, K\}$ **do**
2:     Generate samples $\mathcal{D}_i = \{\tau_{i,j}\}$ by running either $p_i$ or $\pi_{\theta i}$
3:     Fit linear-Gaussian dynamics $p_i(\mathbf{x}_{t+1}|\mathbf{x}_t, \mathbf{u}_t)$ using samples in $\mathcal{D}_i$
4:     Fit linearized global policy $\bar{\pi}_{\theta i}(\mathbf{u}_t|\mathbf{x}_t)$ using samples in $\mathcal{D}_i$
5:     C-step: $p_i \leftarrow \arg\min_{p_i} E_{p_i(\tau)}[\sum_{t=1}^{T} \ell(\mathbf{x}_t, \mathbf{u}_t)]$ such that $D_{\text{KL}}(p_i(\tau)\|\bar{\pi}_{\theta i}(\tau)) \leq \epsilon$
6:     S-step: $\pi_\theta \leftarrow \arg\min_\theta \sum_{t,i,j} D_{\text{KL}}(\pi_\theta(\mathbf{u}_t|\mathbf{x}_{t,i,j})\|p_i(\mathbf{u}_t|\mathbf{x}_{t,i,j}))$ (via supervised learning)
7:     Adjust $\epsilon$ (see Section 4.2)
8: **end for**
---

### B.1  C-Step Details

The C-step solves the following constrained optimization problem:

$$p_i \leftarrow \arg\min_{p_i} E_{p_i(\tau)}\left[\sum_{t=1}^{T} \ell(\mathbf{x}_t, \mathbf{u}_t)\right] \text{ such that } D_{\text{KL}}(p_i(\tau)\|\bar{\pi}_{\theta i}(\tau)) \leq \epsilon.$$

The solution to this problem follows prior work [5], and is reviewed here for completeness. First, the Lagrangian of this problem is given by

$$\mathcal{L}(p_i, \eta) = E_{p_i(\tau)} \left[ \sum_{t=1}^{T} \ell(\mathbf{x}_t, \mathbf{u}_t) \right] + \eta(D_{\mathrm{KL}}(p_i(\tau) \| \bar{\pi}_{\theta i}(\tau)) - \epsilon)$$

$$= \sum_{t=1}^{T} E_{p_i(\mathbf{x}_t, \mathbf{u}_t)}[\ell(\mathbf{x}_t, \mathbf{u}_t) - \eta \log \bar{\pi}_{\theta i}(\mathbf{u}_t | \mathbf{x}_t)] - \eta \mathcal{H}(p(\mathbf{u}_t | \mathbf{x}_t)) - \eta \epsilon,$$

where equality follows from the identity in Appendix A. As discussed in prior work [5], we can minimize this Lagrangian with respect to $p_i$ by solving an LQR problem (assuming a quadratic expansion of $\ell(\mathbf{x}_t, \mathbf{u}_t)$) with a surrogate cost

$$\tilde{\ell}(\mathbf{x}_t, \mathbf{u}_t) = \frac{1}{\eta} \ell(\mathbf{x}_t, \mathbf{u}_t) - \log \bar{\pi}_{\theta i}(\mathbf{u}_t | \mathbf{x}_t).$$

This follows because LQR can be shown to solve the following problem [5]

$$p_i = \arg\min_{p_i} \sum_{t=1}^{T} E_{p_i(\mathbf{x}_t, \mathbf{u}_t)} \left[ \tilde{\ell}(\mathbf{x}_t, \mathbf{u}_t) \right] - \mathcal{H}(p_i(\mathbf{u}_t | \mathbf{x}_t))$$

if we set $p_i(\mathbf{u}_t | \mathbf{x}_t) = \mathcal{N}(\mathbf{K}_t \mathbf{x}_t + \mathbf{k}_t, Q_{\mathbf{u},\mathbf{u}t}^{-1})$, where $\mathbf{K}_t$ and $\mathbf{k}_t$ are the optimal feedback and feedforward terms, respectively, and $Q_{\mathbf{u},\mathbf{u}t}$ is the action component of the Q-function matrix computed by LQR, where the full Q-function is given by

$$Q(\mathbf{x}_t, \mathbf{u}_t) = \frac{1}{2}\mathbf{x}_t^{\mathrm{T}} Q_{\mathbf{x},\mathbf{x}t}\mathbf{x}_t + \frac{1}{2}\mathbf{u}_t^{\mathrm{T}} Q_{\mathbf{u},\mathbf{u}t}\mathbf{u}_t + \mathbf{u}_t^{\mathrm{T}} Q_{\mathbf{u},\mathbf{x}t}\mathbf{x}_t + \mathbf{x}_t^{\mathrm{T}} Q_{\mathbf{x}t} + \mathbf{u}_t^{\mathrm{T}} Q_{\mathbf{u}t}.$$

This maximum entropy LQR solution also directly from the so-called Kalman duality, which describes a connection between LQR and Kalman smoothing.

Once we can minimize the Lagrangian with respect to $p_i$, we can solve the original constrained problem by using dual gradient descent to iteratively adjust the dual variable $\eta$. Since there is only a single dual variable, we can find it very efficiently by using a bracketing line search, exploiting the fact that the dual function is convex.

As discussed in the paper, the dynamics $p_i(\mathbf{x}_{t+1} | \mathbf{x}_t, \mathbf{u}_t)$ are estimated by using samples (drawn from either the local policy or the global policy) and linear regression. Following prior work [5], the dynamics at each step are fitted using linear regression with a Gaussian mixture model prior. This prior incorporates samples from other time steps and previous iterations to allow the regression procedure to use a very small number of sampled trajectories.

## B.2 S-Step Details

The step solves the following optimization problem:

$$\pi_\theta \leftarrow \arg\min_\theta \sum_{t,i,j} D_{\mathrm{KL}}(\pi_\theta(\mathbf{u}_t | \mathbf{x}_{t,i,j}) \| p_i(\mathbf{u}_t | \mathbf{x}_{t,i,j})).$$

Since both $\pi_\theta(\mathbf{u}_t | \mathbf{x}_t) = \mathcal{N}(\mu^\pi(\mathbf{x}_t), \Sigma^\pi(\mathbf{x}_t))$ and $p_i(\mathbf{u}_t | \mathbf{x}_t) = \mathcal{N}(\mathbf{K}_{ti}\mathbf{x}_t + \mathbf{k}_{ti}, \mathbf{C}_{ti})$ are assumed to be conditionally Gaussian, this objective can be rewritten in closed form as

$$\pi_\theta \leftarrow \arg\min_\theta \sum_{t,i,j} \mathrm{tr}[\mathbf{C}_{ti}^{-1}\Sigma^\pi(\mathbf{x}_{t,i,j})] - \log|\Sigma^\pi(\mathbf{x}_{t,i,j})| +$$

$$(\mu^\pi(\mathbf{x}_{t,i,j}) - \mu_{ti}^p(\mathbf{x}_{t,i,j}))\mathbf{C}_{ti}^{-1}(\mu^\pi(\mathbf{x}_{t,i,j}) - \mu_{ti}^p(\mathbf{x}_{t,i,j})).$$

Note that the last term is simply a weighted quadratic cost on the policy mean $\mu^\pi(\mathbf{x}_{t,i,j})$, which lends itself to simple and straightforward optimization using stochastic gradient descent. In our implementation, we use a policy where the covariance $\Sigma^\pi(\mathbf{x}_t)$ is independent of the state $\mathbf{x}_t$, and therefore we can solve for the covariance in closed form, as discussed in prior work [6]. However, in general, the covariance could also be optimized using stochastic gradient descent.

## B.3   Step Size Adjustment

In prior work [8], the step size $\epsilon$ in the local policy optimization is adjusted by considering the difference between the predicted change in the cost of the local policy $p(\mathbf{u}_t|\mathbf{x}_t)$ under the fitted dynamics, and the actual cost obtained when sampling from that policy. The intuition is that, because the linearized dynamics are local, we incur a larger cost the further we deviate from the previous policy. We can adjust the step size by estimating the rate at which the additional cost is incurred and choosing the optimal tradeoff. Let $\ell_{k-1}^{k-1}$ denote the expected cost under the previous local policy $\bar{p}(\mathbf{u}_t|\mathbf{x}_t)$, $\ell_{k-1}^k$ the cost under the current local policy $p(\mathbf{u}_t|\mathbf{x}_t)$ and the previous fitted dynamics (which were estimated using samples from $\bar{p}(\mathbf{u}_t|\mathbf{x}_t)$ and used to optimize $p(\mathbf{u}_t|\mathbf{x}_t)$), and $\ell_k^k$ the cost of the current local policy under the dynamics estimated using samples from $p(\mathbf{u}_t|\mathbf{x}_t)$ itself. Each of these can be computed analytically under the linearized dynamics. We can view the difference $\ell_k^k - \ell_{k-1}^k$ as the additional cost we incur from imperfect dynamics estimation. Previous work suggested modeling the change in cost as a function of $\epsilon$ as following: $\ell_k^k - \ell_{k-1}^{k-1} = a\epsilon^2 + b\epsilon$, where $b$ is the change in cost per unit of KL-divergence, and $a$ is additional cost incurred due to inaccurate dynamics [8]. This model is reasonable because the integral of a quadratic cost under a linear-Gaussian system changes roughly linearly with KL-divergence. The additional cost due to dynamics errors is assumes to scale superlinearly, allowing us to solve for $b$ by looking at the difference $\ell_k^k - \ell_{k-1}^k$ and then solving for a new optimal $\epsilon'$ according to $\epsilon' = -b/2a$, resulting in the update $\epsilon' = \epsilon(\ell_{k-1}^k - \ell_{k-1}^{k-1})/2(\ell_{k-1}^k - \ell_k^k)$.

All of these terms can be computed analytically, since the fitted dynamics, local policies, and linearized global policy $\bar{\pi}_\theta(\mathbf{u}_t|\mathbf{x}_t)$ are all linear-Gaussian. The state-action marginals $p(\mathbf{x}_t, \mathbf{u}_t)$ in linear-Gaussian policies can be computed simply by propagating Gaussian densities forward in time, according to

$$\mu_{\mathbf{x}_t,\mathbf{u}_t} = \begin{bmatrix} \mu_{\mathbf{x}_t} \\ \mathbf{K}_t\mu_{\mathbf{x}_t} + \mathbf{k}_t \end{bmatrix} \qquad \Sigma_{\mathbf{x}_t,\mathbf{u}_t} = \begin{bmatrix} \Sigma_{\mathbf{x}_t} & \Sigma_{\mathbf{x}_t}\mathbf{K}_t^{\mathrm{T}} \\ \mathbf{K}_t\Sigma_{\mathbf{x}_t} & \mathbf{K}_t\Sigma_{\mathbf{x}_t}\mathbf{K}_t^{\mathrm{T}} + \mathbf{C}_t \end{bmatrix}$$

$$\mu_{\mathbf{x}_{t+1}} = f_t\mu_{\mathbf{x}_t,\mathbf{u}_t} + f_{ct} \qquad \Sigma_{\mathbf{x}_{t+1}} = f_t\Sigma_{\mathbf{x}_t,\mathbf{u}_t}f_t^{\mathrm{T}} + \mathbf{F}_t$$

where we have $p(\mathbf{x}_{t+1}|\mathbf{x}_t, \mathbf{u}_t) = \mathcal{N}(f_t(\mathbf{x}_t, \mathbf{u}_t)^{\mathrm{T}} + f_{ct}, \mathbf{F}_t)$ and $p(\mathbf{u}_t|\mathbf{x}_t) = \mathcal{N}(\mathbf{K}_t\mathbf{x}_t + \mathbf{k}_t, \mathbf{C}_t)$, and then we can estimate the expectation of the cost at time $t$ simply by integrating the quadratic cost under the Gaussian state-action marginals.

In MDGPS, we propose to use two step size adjustment rules. The first rule simply adapts the previous method to the case where we constrain the new local policy $p(\mathbf{u}_t|\mathbf{x}_t)$ against the global policy $\pi_\theta(\mathbf{u}_t|\mathbf{x}_t)$, instead of the previous local policy $\bar{p}(\mathbf{u}_t|\mathbf{x}_t)$. In this case, we simply replace $\ell_{k-1}^{k-1}$ with the expected cost under the previous global policy, given by $\ell_{k-1}^{k-1,\pi}$, obtained using its linearization $\bar{\pi}_\theta(\mathbf{u}_t|\mathbf{x}_t)$. We call this the "classic" step size: $\epsilon' = \epsilon(\ell_{k-1}^k - \ell_{k-1}^{k-1,\pi})/2(\ell_{k-1}^k - \ell_k^k)$.

However, we can also incorporate intuition from the bound in Section 4.1 to obtain a more conservative step adjustment that reduces $\epsilon$ not only when the obtained local policy improvement doesn't meet expectations, but also when we detect that the global policy is unable to reproduce the behavior of the local policy. In this case, reducing $\epsilon$ reduces the KL-divergence between the global and local policies which, as shown in the previous section, tightens the bound on the global policy return. As mentioned previously, directly optimizing the bound tends to perform poorly because the bound is quite loose. However, if we estimate the cost of the global policy using its linearization, we can instead adjust the step size based on a simple model of *global* policy cost. We use the same model for the change in cost, given by $\ell_k^{k,\pi} - \ell_{k-1}^{k-1,\pi} = a\epsilon^2 + b\epsilon$. However, for the term $\ell_k^k$, which reflects the actual cost of the new policy, we instead use the cost of the new global policy $\ell_k^{k,\pi}$, so that $a$ now models the additional loss due to *both* inaccurate dynamics and inaccurate projection: if $\ell_k^{k,\pi}$ is much worse than $\ell_{k-1}^k$, then either the dynamics were too local, or S-step failed to match the performance of the local policies. In either case, we decrease the step size.[4] As before, we can solve for the new step size $\epsilon'$ according to $\epsilon' = \epsilon(\ell_{k-1}^k - \ell_{k-1}^{k-1,\pi})/2(\ell_{k-1}^k - \ell_k^{k,\pi})$. We call this the "global" step size.

## C  Global Policy Cost Bounds

In this appendix, we prove the bound on the policy cost discussed in Section 4.1. The proof combines the earlier results from Ross et al. [15] and Schulman et al. [17], and extends them to the case of finite-horizon episodic tasks.

### C.1  Policy State Distribution Bound

We begin by proving Lemma 4.1, which we restate below with slightly simplified notation, replacing $\pi_\theta$ by $q$:

**Lemma C.1** *Let* $\epsilon_t = \max_{\mathbf{x}_t} D_{KL}(p(\mathbf{u}_t|\mathbf{x}_t)\|q(\mathbf{u}_t|\mathbf{x}_t))$. *Then* $D_{TV}(p(\mathbf{x}_t)\|q(\mathbf{x}_t)) \leq 2\sum_{t=1}^{T} \sqrt{2\epsilon_t}$.

The proof first requires introducing a lemma that relates the total variation divergence $\max_{\mathbf{x}_t} \|p(\mathbf{u}_t|\mathbf{x}_t) - q(\mathbf{u}_t|\mathbf{x}_t)\|_1$ between two policies to the probability that the policies will take the same action in a discrete setting (extensions to the continuous setting are also possible):

**Lemma C.2** *Assume that* $\max_{\mathbf{x}_t} \|p(\mathbf{u}_t|\mathbf{x}_t) - q(\mathbf{u}_t|\mathbf{x}_t)\|_1 \leq \sqrt{2\epsilon_t}$, *then the probability that $p$ and $q$ take the same action at time step $t$ is* $1 - \sqrt{2\epsilon_t}$.

The proof for this lemma was presented by Schulman et al. [17]. We can use it to bound the state distribution difference as following. First, we are acting according to $p(\mathbf{u}_t|\mathbf{x}_t)$, the probability that the same action would have been taken by $q(\mathbf{u}_t|\mathbf{x}_t)$, based on Lemma C.2, is $(1 - \sqrt{2\epsilon_t})$, so the probability that all actions up to time $t$ would have been taken by $q(\mathbf{u}_t|\mathbf{x}_t)$ is given by $\prod_{t'=1}^{t}(1 - \sqrt{2\epsilon_{t'}})$. We can therefore express the state distribution $p(\mathbf{x}_t)$ as

$$p(\mathbf{x}_t) = \left[\prod_{t'=1}^{t}(1 - \sqrt{2\epsilon_{t'}})\right] q(\mathbf{x}_t) + \left(1 - \prod_{t'=1}^{t}(1 - \sqrt{2\epsilon_{t'}})\right)\tilde{p}(\mathbf{x}_t)$$

$$= \left[\prod_{t'=1}^{t}(1 - \sqrt{2\epsilon_{t'}})\right][q(\mathbf{x}_t) - \tilde{p}(\mathbf{x}_t)] + \tilde{p}(\mathbf{x}_t),$$

where $\tilde{p}(\mathbf{x}_t)$ is some other distribution. In order to bound $D_{TV}(p(\mathbf{x}_t)\|q(\mathbf{x}_t)) = \|p(\mathbf{x}_t) - q(\mathbf{x}_t)\|_1$, we can substitute this equation into $\|p(\mathbf{x}_t) - q(\mathbf{x}_t)\|_1$ to get

$$\|p(\mathbf{x}_t) - q(\mathbf{x}_t)\|_1 = \left\|\left[\prod_{t'=1}^{t}(1 - \sqrt{2\epsilon_{t'}})\right][q(\mathbf{x}_t) - \tilde{p}(\mathbf{x}_t)] + \tilde{p}(\mathbf{x}_t) - q(\mathbf{x}_t)\right\|$$

$$= \left\|\left[1 - \prod_{t'=1}^{t}(1 - \sqrt{2\epsilon_{t'}})\right][q(\mathbf{x}_t) - \tilde{p}(\mathbf{x}_t)]\right\|$$

$$= \left[1 - \prod_{t'=1}^{t}(1 - \sqrt{2\epsilon_{t'}})\right]\|q(\mathbf{x}_t) - \tilde{p}(\mathbf{x}_t)\|$$

$$\leq 2\left[1 - \prod_{t'=1}^{t}(1 - \sqrt{2\epsilon_{t'}})\right],$$

where the last inequality comes from the fact that $\|q(\mathbf{x}_t) - \tilde{p}(\mathbf{x}_t)\| \leq 2$ for discrete distributions. With continuous densities, we could extend the result by taking the limit of an infinitely fine discretization. Next, we note that

$$\prod_{t'=1}^{t}(1 - \sqrt{2\epsilon_{t'}}) \geq 1 - \sum_{t'}\sqrt{2\epsilon_{t'}},$$

and therefore we have

$$\|p(\mathbf{x}_t) - q(\mathbf{x}_t)\|_1 \leq 2\sum_{t'=1}^{t}\sqrt{2\epsilon_{t'}}$$

This completes the proof.

## C.2 Total Policy Cost Bound

In this appendix, we use the result above to prove Lemma 4.2. This result is based on Ross et al. [15], but extends the proof to the case of time-varying finite-horizon systems. We first restate the lemma under the same notation as the previous appendix:

**Lemma C.3** *If $D_{TV}(p(\mathbf{x}_t) \| \pi_\theta(\mathbf{x}_t)) \leq 2 \sum_{t=1}^{T} \sqrt{2\epsilon_t}$, then we can bound the total cost of $\pi_\theta$ as*

$$\sum_{t=1}^{T} E_{\pi_\theta(\mathbf{x}_t, \mathbf{u}_t)}[\ell(\mathbf{x}_t, \mathbf{u}_t)] \leq \sum_{t=1}^{T} \left[ E_{p(\mathbf{x}_t, \mathbf{u}_t)}[\ell(\mathbf{x}_t, \mathbf{u}_t)] + 2\sqrt{\epsilon_t} Q_{max,t} \right],$$

*where $Q_{max,t} = \sum_{t'=t}^{T} \max_{\mathbf{x}_{t'}, \mathbf{u}_{t'}} \ell(\mathbf{x}_{t'}, \mathbf{u}_{t'})$, the maximum total cost from time $t$ to $T$.*

We bound the cost of $q$ at time step $t$ according to

$$
\begin{aligned}
E_{q(\mathbf{x}_t, \mathbf{u}_t)}[\ell(\mathbf{x}_t, \mathbf{u}_t)] &= \langle q(\mathbf{x}_t, \mathbf{u}_t), \ell(\mathbf{x}_t, \mathbf{u}_t) \rangle \\
&= \langle q(\mathbf{x}_t, \mathbf{u}_t) - p(\mathbf{x}_t)q(\mathbf{u}_t|\mathbf{x}_t), \ell(\mathbf{x}_t, \mathbf{u}_t) \rangle + \langle p(\mathbf{x}_t)q(\mathbf{u}_t|\mathbf{x}_t), \ell(\mathbf{x}_t, \mathbf{u}_t) \rangle \\
&= \langle q(\mathbf{u}_t|\mathbf{x}_t)[q(\mathbf{x}_t) - p(\mathbf{x}_t)], \ell(\mathbf{x}_t, \mathbf{u}_t) \rangle + \langle p(\mathbf{x}_t)[q(\mathbf{u}_t|\mathbf{x}_t) - p(\mathbf{u}_t|\mathbf{x}_t)], \ell(\mathbf{x}_t, \mathbf{u}_t) \rangle + E_{p(\mathbf{x}_t, \mathbf{u}_t)}[\ell(\mathbf{x}_t, \mathbf{u}_t)] \\
&\leq E_{p(\mathbf{x}_t, \mathbf{u}_t)}[\ell(\mathbf{x}_t, \mathbf{u}_t)] + \|q(\mathbf{x}_t) - p(\mathbf{x}_t)\|_1 \max_{\mathbf{x}_t, \mathbf{u}_t} \ell(\mathbf{x}_t, \mathbf{u}_t) + \|q(\mathbf{u}_t|\mathbf{x}_t) - p(\mathbf{u}_t|\mathbf{x}_t)\|_1 \max_{\mathbf{x}_t, \mathbf{u}_t} \ell(\mathbf{x}_t, \mathbf{u}_t) \\
&\leq E_{p(\mathbf{x}_t, \mathbf{u}_t)}[\ell(\mathbf{x}_t, \mathbf{u}_t)] + \max_{\mathbf{x}_t, \mathbf{u}_t} \ell(\mathbf{x}_t, \mathbf{u}_t) \sqrt{2\epsilon_t} + 2 \max_{\mathbf{x}_t, \mathbf{u}_t} \ell(\mathbf{x}_t, \mathbf{u}_t) \sum_{t'=1}^{t} \sqrt{2\epsilon_{t'}}
\end{aligned}
$$

If we add up the above quantity over all time $t$, we get

$$\sum_{t=1}^{T} E_{q(\mathbf{x}_t, \mathbf{u}_t)}[\ell(\mathbf{x}_t, \mathbf{u}_t)] \leq \sum_{t=1}^{T} E_{p(\mathbf{x}_t, \mathbf{u}_t)}[\ell(\mathbf{x}_t, \mathbf{u}_t)] + \sum_{t=1}^{T} \sqrt{2\epsilon_t} \max_{\mathbf{x}_t, \mathbf{u}_t} \ell(\mathbf{x}_t, \mathbf{u}_t) + 2 \sum_{t=1}^{T} \max_{\mathbf{x}_t, \mathbf{u}_t} \ell(\mathbf{x}_t, \mathbf{u}_t) \sum_{t'=1}^{t} \sqrt{2\epsilon'_t}$$

which we can rewrite as

$$\sum_{t=1}^{T} E_{q(\mathbf{x}_t, \mathbf{u}_t)}[\ell(\mathbf{x}_t, \mathbf{u}_t)] \leq \sum_{t=1}^{T} \left[ E_{p(\mathbf{x}_t, \mathbf{u}_t)}[\ell(\mathbf{x}_t, \mathbf{u}_t)] + \sqrt{2\epsilon_t} \max_{\mathbf{x}_t, \mathbf{u}_t} \ell(\mathbf{x}_t, \mathbf{u}_t) + 2\sqrt{2\epsilon_t} Q_{max,t} \right]$$

where $Q_{max,t} = \sum_{t'=t}^{T} \max_{\mathbf{x}_{t'}, \mathbf{u}_{t'}} \ell(\mathbf{x}_{t'}, \mathbf{u}_{t'})$.

## Footnotes

[4] Although we showed before that the discrepancy depends on $\sum_{t=1}^T \sqrt{2\epsilon_t}$, here we use $\epsilon^2$. This is a simplification, but the net result is the same: when the global policy is worse than expected, $\epsilon$ is reduced.