[Reviews · NeurIPS 2016]

Reviewer 1

Summary

The authors propose a new, simplified approach to guided policy search. Guided policy search consists in first improving a set of local policies starting from different initial states, second using supervised learning to "compile" them into a global policy, and then iterating those two steps towards convergence of local and global policies. In the previous method, a constraint was added through a surrogate cost function so that the local policies and global policies converge to each other. In this new approach, the surrogate cost function is removed and the process is simpler, based on minimizing the KL divergence between local and global policies. This simplification was made possible by recognizing the similarity between guided policy search and mirror descent. A very preliminary experimental study shows some advantages of the new approach.

Qualitative Assessment

Despite my initial lack of knowledge about guided policy search methods, which makes the paper hard to read for me, I believe this is a strong paper presenting state-of-the-art research. However, there are a few issues that I'm listing hereafter by order of decreasing importance. My main point is with the way Algorithm 2 is written. In contrast to Algorithm 1, the C-step of Algorithm 2 requires to already have a global policy \pi_\theta so as to compute the KL divergence. Nothing is said about providing an initial policy \pi_\theta_0. Did I miss something here? Or shouldn't you split step 2 into a step 2 and a step 4 where you perform the re-optimization of local policies, as in Algo 1? If I'm right, aren't there some constraints on the way \pi_\theta is initialized? In any case, this should be clarified. The experimental evaluation is rather weak: - the description of the experiments is sketchy, one cannot reproduce them out of what is described (which simulator?, position of the holes?, size of the arm?, etc.) - the dimensions for x_t and u_t are not given - results are averaged over 3 runs only. Statistical significance is not checked. Worse than that, I doubt the validity of the success rates: how can you get a success rate of 62.50% over three runs with 4 holes? It means you succeeded 7,5 times (!). Something is not specified here (are there several instances of peg insertion for the same hole over one run?). - computational cost is not mentioned: it is high, to explain why averaging over 3 runs only? A comparison of computational cost and sample efficiency with respect to previous guided policy search methods or simpler baselines such as TRPO looks mandatory - the authors put scalability forward, but their policy is a 2 layers network with 40 neurons each, which is rather small. This should be discussed. On one side, the paper is difficult and long. A good deal of the technical part of the paper is rejected into the supplementary material, and some sections of the paper can hardly be read without reading this supplementary part. I must say I didn't check many technical details in the methods nor the proofs in the supplementary material. On the other side, I didn't like much Section 4.2, which tries to justify a rather elaborated policy for step size selection, but the message is ruined by footnote 3, which states that finally using a simple rule of thumb is enough. So I would suggest the authors to remove Section 4.2 and use the resulting free space to clarify the experimental section and the methods. Also, wouldn't Section 3.2 find a better place into Section 4? It is never specified what is the domain of x_t and u_t. Are they in R^n? In a closed domain of R^n? More local points: Lines 55-58, this should be written more clearly. The first sentence states that policy gradient methods can be applied to huge policies (citing recent deep methods), and the second states that they are limited to small policies (citing older work). You may rather insist on number of episodes for training, as done later on. Maybe you should not use casual writing such as "we'll, we've, we don't"... in scientific papers. I just found one typo: In Algo 1, and each p(u_t|x_t) should be each p_i(u_t|x_t)

Confidence in this Review

1-Less confident (might not have understood significant parts)


Reviewer 2

Summary

In this paper the author proposed a policy search approach that 1) locally improves the performance of the cumulative cost minimization problem while following the actions induced by previous policy, and 2) generalize the local policies to various initial conditions via supervised learning. This methodology is very similar to the existing methods in the guided policy search literature.

Qualitative Assessment

My major concern to this paper is its novelty and level of contribution. While the authors clearly notice the work by Sergey et. al. in guided policy search (GPS) from their previous publications in 2013 and 2014, they have not clearly differentiated their methods with the existing literature. Based on my understanding to GPS, I cannot see any major differences between the proposed method and the standard GPS algorithm. Results in appendix B is very similar to the result in the appendix of Sergey's 2014 NIPS GPS paper (http://homes.cs.washington.edu/~svlevine/papers/mfcgps.pdf). Also the results in appendix C on the relationship between KL divergence and total variation are straightforward (see Section 3 of Schulman's TRPO paper: https://arxiv.org/abs/1502.05477). Finally, the notion of mirror descent is very confusing here. Are the authors trying to bound the performance of the algorithms (in terms of regret analysis or sample complexity) and treating the KL cost as the Bregman divergence of an Entropic regularizer (similar to i.e. the regularizer of Exp3 in online learning)?? This is unclear to me, for example, are their any conjugate cost analysis of GPS based on the Fenchel duality of the negative entropy?

Confidence in this Review

3-Expert (read the paper in detail, know the area, quite certain of my opinion)


Reviewer 3

Summary

The paper proposes to view guided policy search as an approximate mirror descent. Based on this interpretation, the authors propose a new version of guided policy search, which corresponds to mirror descent when the dynamics is linear and the cost is quadratic. When the dynamics is nonlinear, the projection of this algorithm is done approximately. In that case, they provide a bound on the costs of the local and global policies, which may be used for stepsize selection in the algorithm. Finally, the experimental results show that the new algorithm has similar performance to BADMM, a previous version of guided policy search.

Qualitative Assessment

The paper is well-written and clear. The contribution of this paper builds on previous work on guided policy search. The new proposed algorithm has the advantage of having less hyperparameters than BADMM. It seems to provide similar performance results to BADMM at the cost of slower convergence. In the experimental results, the authors compare the performance of BADMM with those of the local policies (used to teach the global policy). However, are we not only interested in the global policy at the end? Why is that comparison interesting? Besides, the experimental results are averaged only over 3 runs, which seems to be very small. As the experiments are done in a simulated environment, I think more runs should have been performed. Typos: paper: l.4 of Algo.1: p_i(\mathbf u_t | \mathbf x_t) l.52: \tau = (x_1, u_1, \ldots, x_T, u_T) l.83: a factor 1/N is missing in J(\theta) l.128: constrain l.168: Algorithm 3 l.191: missing parenthesis for D_{KL} supplementary material: l.382: Algoritm 3

Confidence in this Review

2-Confident (read it all; understood it all reasonably well)


Reviewer 4

Summary

The paper proposes a novel method for guided policy search that uses supervised learning algorithm to mimic a policy provided by a teacher or a trajectory optimizer. In comparison to previous guided policy search algorithms, the proposed method has convergence guarantees for linear and convex systems, and bounded errors for non-linear systems. The proposed method is also simpler to tune and use. Another contribution of the paper is the interpretation of guided policy search algorithms as an approximate version of mirror descent. Empirical verification on several difficult robotics tasks in simulation show that the proposed method performs better than or as well as existing methods.

Qualitative Assessment

The state spaces of the three robotic tasks are more or less clear from the description in Section 6, but what are the available actions for each of them?

Confidence in this Review

1-Less confident (might not have understood significant parts)


Reviewer 5

Summary

This paper introduce a new guided police search called mirror descent guided policy search and proposed implementations for model-base and model-free variants. The new method sheds light on the study of guided police search and try to provide theoretical understanding of guided policy search method. Also, the authors provide empirical demonstrations to evaluate their method.

Qualitative Assessment

Overall, this paper is intriguing and well organized. The authors summarized the classical guided policy search method and then proposed the new methods. Also, the authors show some theoretical and empirical results. But I have some suggestions and comments. 1. The title is slightly misleading. When I saw the title and read through the abstract, I thought this paper focused on the theoretical study of guided policy search methods. However, after reading through all the sections, I realize that this paper aims to design a new guided policy search algorithm and mainly focused on the practical applications of the new algorithm. Here are some suggestions. Could the authors change the title into "Guided Policy Search via Approximate Mirror Descent". In the abstract, could the authors emphasize on the introduction to the new algorithm via approximate mirror descent instead of "be interpreted as ..."? 2. The theoretical analysis of this paper is not deep enough. That's why I lower the novelty and potential impact scores. The authors only provide a bound about the total cost of \pi_{\theta}. Therefore, I have some suggestions: 2a) Could the authors provide some sample complexity or convergence rate for the new algorithms? This will make the work more useful and novel in terms of theoretical understanding. 2b) In section 3.2 the author mentioned "The main distinction between the proposed method and prior guided policy search methods is that the constraint D \leq \epislon is enforced on the local policies at each iteration, while in prior methods, this constraint is iteratively enforced via a dual descent procedure over multiple iterations." Could the authors theoretically study the difference between proposed method and prior approaches? For example, how is the constraints at each iteration different from constraints on multiple iterations. How it slows the algorithm theoretically in the sense of convergence rate, etc. Unfortunately, the authors only provide some empirical discussion. I think this will be a very important result since it tries to understand the guided policy search in a deeper level. 2c) Section 4.2 Step size selection is not so important. The authors could put most of its contents in the Appendix 3. For the experimental results, the authors provide three domains. I think due to the space limit, the authors only provide minimum details about the experimental setting. Could the authors provide some details about the experiments in the appendix? For example, hyper-parameter tuning, experimental platform and so on. In all, this is an interesting work. It provides a new algorithm for guided policy search study. However, there are some improvements for this paper, which will make this paper better and more impressive.

Confidence in this Review

2-Confident (read it all; understood it all reasonably well)


Reviewer 6

Summary

This paper suggests a new method of guided policy search, called mirror descent guided policy search (MDGPS), which is influenced by mirror descent method. Rather to modify the loss function to constraints that next local policies are trained not too far from the global policy, this method uses KL-divergence to constraint the distance between the local and global policy. The paper also explains the case that the global policy is nonlinear and dynamics is unknown by using linear-Gaussian dynamics. The paper provides the theoretical guarantees of that the upper bound of the expected sum of loss of the global policy using that of the local policy and the step size. Based on this result, the paper also suggests a new method to select the step size using the global policy. They evaluate their method on several domains including synthetic dataset and real robotic domains, and shows that their method outperforms the previous GPS method based on BADMM in a view of success rate of the problem.

Qualitative Assessment

This paper is nice to read, and provides good theoretical analysis for a new suggesting algorithm. By using KL-divergence, this paper constructs an algorithm with robust theoretical analysis and reduce the model parameters. However, borrowing more complex modeling also makes the drawbacks and thus this algorithm runs slower than previous methods. It is little disappointed, however, that the performance of MDGPS shows the similar result with the BADMM based GPS as shown in Figure 1, because MDGPS consumes more computations and more complex models. I also have a question why the graph in Figure 1 seems little noisy, and there is no standard error information on Table 1. Maybe more number of experiments makes the result more concrete. As a minor thing, standard error plot on Figure 1. seems little wrong, for example in the left side plot, standard error of ‘Off policy, Classic Step’ plot is on wrong place.

Confidence in this Review

1-Less confident (might not have understood significant parts)